# Unsupervised Clustering of Forest Response to Drought Stress in Zululand Region, South Africa

**Sifiso Xulu [1,2,*] , Kabir Peerbhay [1] , Michael Gebreslasie [1] and Riyad Ismail [1]**

[1]  School of Agricultural, Earth and Environmental Sciences, University of KwaZulu-Natal, Westville Campus, Durban 4000, South Africa
[2]  Department of Geography and Environmental Studies, University of Zululand, KwaDlangezwa 3886, South Africa
*  Correspondence: xulusi@unizulu.ac.za; Tel.: +27-035-902-6331

**Abstract:** Drought limits the production of plantation forests, notably in the drought-prone Zululand region of South Africa. During the last 40 years, the country has faced a series of severe droughts, however that of 2015 stands out as the most extreme and prolonged. The 2015 drought impaired forest productivity and led to widespread tree mortality in this region, but the identification of tree response to drought stress remains uncertain because of its spatial variability. To address this problem, a method that can capture drought patterns and identify trees with similar reactions to drought stress is desired. This could improve the accuracy of detecting trees suffering from drought stress which is key for forest management planning. In this study, we aimed to evaluate the utility of unsupervised mapping approaches in compartments of *Eucalyptus* trees with similar drought characteristics based on the Normalized Difference Water Index (NDWI) and to demonstrate the value of cloud-based Google Earth Engine (GEE) resources for rapid landscape drought monitoring. Our results showed that calculating distances between pixels using three different matrices (Random Forest (RF) proximity, Euclidean and Manhattan) can accurately detect similarities within a dataset. The RF proximity matrix produced the best measures, which were clustered using Wards hierarchical clustering to detect drought with the highest overall accuracy of 87.7%, followed by Manhattan (85.9%) and Euclidean similarity measures (79.9%), with user and producer results between 84.2% to 91.2%, 42.8% to 98.2% and 37.2% to 94.7%, respectively. These results confirm the value of the RF proximity matrix and underscore the capability of automatic unsupervised mapping approaches for monitoring drought stress in tree plantations, as well as the value of using GEE for providing cost effective datasets to resource stricken countries.

**Keywords:** Unsupervised Random Forest clustering; drought; plantation forests; normalized difference water index; Google Earth Engine

---

## 1. Introduction

The commercial forestry sector plays an important role in the economic development of South Africa as it contributes a net revenue of approximately R31 billion (USD $918,092,058) to the gross domestic product and employs more than 200,000 of the national labour force [1]. However, the most productive Zululand forestry region along the north-east coast of the country has been hit by a series of severe droughts, causing a corresponding decline in *Eucalyptus* productivity [2], and widespread tree mortality [3]. For example, downward growth patterns were highest during the recent 2015 drought, where reductions ranging from 35% to 40% were recorded around the Mtubatuba area [4]. This phenomenon is characterized by insufficient soil moisture to support tree growth during their reproductive phase [5] and is the natural outcome of anomalous precipitation deficits and high

temperatures [6]. In response to this strain, the Institute for Commercial Forestry Research sourced a variety of *Corymbia* material from Australia and established two × *Corymbia* hybrid interaction trials within the Zululand region in 2013 [4]. An important feature of the *Corymbia* hybrids include increased tolerance to drier drought conditions and, particularly during the 2015 drought event, they were found to perform exceptionally well compared to *Eucalyptus* in KwaMbonambi [4]. Some forestry plantations in this region are projected to suffer from frequent and severe droughts [7,8] and this will impose new challenges for timber production. Moreover, the impact of drought on forest trees essentially depends on its duration, frequency and magnitude, but also varies with species' sensitivity to drought stress [9]. Beyond South Africa, many studies have also reported widespread and accelerated forest mortality due to droughts, ranging from partial but consistent increases in background mortality rates [10] to large-scale die-offs [11–14]. Exploring how drought impacts on plantation trees is therefore vital to improve our understanding of the effects of ongoing climatic changes on these economically valuable resources.

Over the past four decades, South Africa has faced a series of economically damaging droughts, however the 2015 drought stands out as the most extreme [15], costing farmers losses of up to R10m (USD $698,850) [16]. During this period, the country received a record-setting (403 mm) annual rainfall since the South African Weather Service began collecting rainfall data in 1904 [17]; a record lower than the historical mean annual rainfall of 739 mm. Also, the 2015 drought came on the back of three successive years (2014–2016) of lower-than-normal rainfall, making it the most severe and prolonged drought since the 1940s [17]. This event was associated with the strongest El Niño event of climate records [15]. Such conditions prevail typically during the mature El Niño–Southern Oscillation phenomenon when the central and eastern tropical Pacific and the Indian Ocean are warmer than average [18]. The strong El Niño signal in the semi-arid regions of southern Africa is well-known to bring drier conditions which adversely affect vegetation productivity [19]. The catastrophic character of the 2015 drought led to the official declaration of a disaster in terms of the country's Disaster Management Act of 2002 [20], which seeks to reduce drought related risk and offer agricultural risk insurance which is aimed at enhancing the income of farmers and producers who are most vulnerable to losses as a result of drought. Its pronounced effects are reflected in Xulu et al. [2]. *Eucalyptus* trees showed the most extreme response, resulting in extensive tree dieback [3]. Therefore, identifying varieties that are resistant to drought and planting them in drought-prone areas may increase the resilience of plantations. Again, accurate detection of drought stress symptoms or drought damaged trees allow for rapidly estimating crop yields and assists in strategic forest protection decisions.

However, the detection of drought is complex [21] and is different from other forest-damaging agents (such as fire) in several respects. For example, under similar drought stress conditions, different species [22], and even individual trees [3], can experience differential rates of decline and mortality levels, which can cause variations in competitive capacity and affect the species composition of forests [23–25]. Secondly, its consequences can be experienced at relatively broad geographic scales and simultaneously across landscapes [26]. Thirdly, drought stimulates other forest disturbances such as fire [27] and insect outbreaks [28], which further impair tree productivity, sometimes leading to tree mortality [29,30]. Lastly, it is challenging to determine the onset and cessation of drought episodes since vegetation responds in different ways [31], making it difficult to evaluate drought using costly and spatially restricted field-based measurements [32]. For example, Crous et al. [3] found that the dying *Eucalyptus grandis* × *Eucalyptus urophylla* clone was more susceptible to drier drought conditions compared to two co-occurring healthy compartments of the same clone. These complex responses present additional difficulties as the outcomes may not be directly comparable over a large area and the severity of impact can vary markedly [33]. Fortunately, drought characteristics are directly recordable using optical sensors [34], making remote sensing an ideal tool for repeated monitoring and over large areal extents.

Various methods, such as high-density time series analysis based on remote sensing indices, have been used to detect the impact of drought due to remotely sensed data being consistent, flexible

and more spatially continuous than other methods (i.e., field surveys) [35]. Hitherto, numerous studies have effectively characterized droughts through simplified drought indices such as the standardized precipitation index (SPI) at various timescales [36,37]. Such an index allows the quantification of climate–vegetation anomalies in terms of magnitude and intensity, duration and spatial scale, consequently enabling the evaluation of drought's impact on vegetation [38]. Because trees exhibit varying sensitivity to drought stress [9], the analysis of vegetation indices should reveal forest response to drought over spatial and temporal scales. The response of forest trees to drought has been widely explored using the ratios of the near-infrared (NIR), red and short-wave infrared (SWIR) bands and their various combinations. Of these, the indices that are based on the NIR and SWIR bands have shown to have the greatest sensitivity to drought stress [39].

For example, Xulu et al. [2] used normalized difference infrared index (NDII) and normalized difference vegetation index (NDVI) to characterize *Eucalyptus'* response to the recent intense drought in South Africa and found the NDII to show the greatest sensitivity to drought stress. While these studies utilized supervised approaches to detect drought, unsupervised mapping methods have also shown success in delineating drought areas with similar characteristics [40]. Unlike the supervised methods, unsupervised learning is capable of discovering patterns in the data on its own. For example, Santos [41] successfully applied principal component analysis (a statistical procedure used to emphasize the variations and patterns in a dataset) and *k*-means (an unsupervised clustering method used to classify unlabeled data based on similarities) to explore spatial and temporal patterns of droughts in Portugal based on SPI, which is the widely used meteorological drought index, computed as a difference of precipitation from the mean for a specified time period divided by the standard deviation where the mean and standard deviation are determined from past records [42]. Rahmat et al. [43] also segmented the Victoria catchment in Australia into six homogenous clusters subject to similar SPI characteristics using hierarchical clustering and SPI. More recently, Xie et al. [44] examined drought characteristics in the Xinjiang province of China using *k*-means clustering and SPI.

In this study, we investigate the utility of a novel Random Forest (RF) unsupervised mapping approach using the proximity matrix for detecting anomalous drought stressed forest compartments in KwaZulu-Natal, South Africa. We used a remotely-sensed normalized difference water index (NDWI) as the basis for the classification exercise. The proximity matrix produces non-linear distances between unlabeled pixels to build patterns within a dataset. These proximity scores then provide an unsupervised automated methodology for spatial clustering [45]. This study provides an exciting opportunity to advance our knowledge of mapping drought-affected compartments using unsupervised RF clustering analysis. It provides a cost-effective method to separate drought-affected from non-affected plantation trees based on remotely sensed data and cloud-based Google Earth Engine (GEE) resources. The results of this study could add value in the identification of drought-affected plantations so that alternative trees that are suited to drought prone sites can be used. The methods applied here can also be tested in other plantations or areas affected by droughts.

## 2. Materials and Methods

### 2.1. Study Area

Our spatial area of analysis is the 20,000 ha plantation forest in KwaMbonambi, which is located 30 km north of the town of Richards Bay along the north-east coast of KwaZulu-Natal (Figure 1). The growing stock mainly comprise 6–14-year-old *Eucalyptus grandis* W. Hill ex Maiden (*E. grandis*) (2%), *E. grandis* × *Eucalyptus camaldulensis* Dehnh. (*E. gxc*) (2%) and *E. grandis* × *Eucalyptus urophylla* S.T. Blake (*E. gxu*) (96%) hybrid clones. *E. gxu* is more drought-sensitive than other hybrid clones in this area. The plantations are managed by Sappi, a large South African pulp and paper company with a global reach. The compartments are relatively uniform in terms of canopy cover with a tree density of 1667 trees ha$^{-1}$, and the average tree height for most compartments is 14.75 m. The area has a subtropical climate, with mean minimum temperatures of 16 °C around June–August and

mean maximum of 27 °C during November–March [46]. The mean annual rainfall varies from 739 to 1219 mm, which is highly seasonal, peaking between November and February [47]; the mean potential evapotranspiration is commonly in the range of 1100 to 1772 mm [48]. The landscape of KwaMbonambi consists of Quaternary alluvial sediments of clay sands of aeolian deposits [49] and soil with varying amounts of organic matter [50] at an elevation of 30 to 100 m above sea level. The high penetrability of the soils permits rapid leaching of their nutrients due to the high rainfall [49]. These conditions are favorable for fast-growing *Eucalyptus* plantations [51].

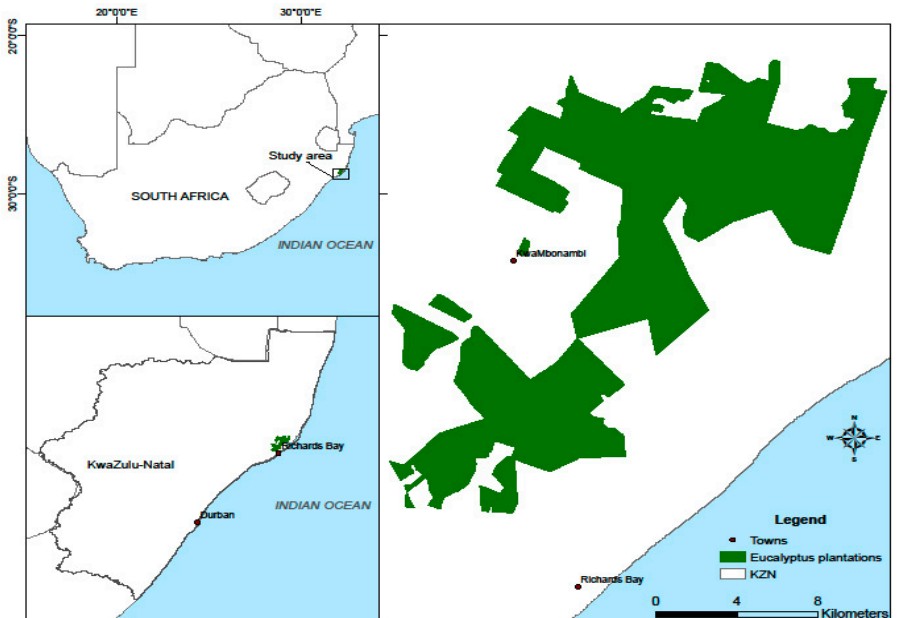

**Figure 1.** Location of the study area in KwaMbonambi, South Africa.

## 2.2. Evaluation of Relative Water Content

The vegetation index used in assembling clusters of compartments with similar responses to drought is the 30 m resolution Landsat-derived normalized difference water index (NDWI); the index represents the ratio of the difference between the near-infrared (NIR; 0.76–0.90 μm) and the shortwave infrared band (SWIR; 1.55–1.75 μm) reflectance over the combined reflectance in these two parts of the spectrum [52], as illustrated in Equation (1).

$$\text{NDWI} = \frac{\text{NIR} - \text{SWIR}}{\text{NIR} + \text{SWIR}} \tag{1}$$

The NDWI ranges from 0 to 1, depending on the water content in the plants, where high NDWI values correspond to high plant water content and low values correspond to low water content in the vegetation. So, the NDWI values are expected to decrease during the period of water stress—that is 2015 in our case.

The data were extracted at no cost and processed using the JavaScript code editor in the GEE platform (https://earthengine.google.com/, Mountain View, CA, USA), which enables parallel computing and extensive data processing. The SWIR reflectance is very sensitive to variations in both leaf water content and the spongy mesophyll structure in the forest canopy, whereas the NIR reflectance is affected by internal leaf structure and leaf dry matter content, but not by water content [52]. The composite signals of the NIR and the SWIR bands removes variants introduced by internal leaf structure, thereby augmenting the accuracy of detecting water content from trees [53]. The SWIR sensitivity to water stress has rendered it as a reliable surrogate for monitoring drought-induced forest disturbance [54]. The cloudy dates were filtered to obtain images captured under cloudless conditions and processed using the JavaScript code editor in the GEE environment, resulting in one to two images per month.

A monthly mean time series of NDWI (2013–2017) for a total of 383 forest compartments was retrieved from the GEE environment. The NDWI values were averaged for the entire compartment individually, and the mean monthly NDWI was used for analysis. The compartments ranged from 0,3 ha to 51 ha in extent. Polygons for all drought stressed (*n* = 57) and non-stressed (*n* = 326) compartments covering the study area of almost 20,000 ha were obtained from Sappi's forest management database.

### 2.3. Calculating Distance Matrices and Unsupervised Clustering

In principle, a cluster analysis is fundamentally linked to determining the similarity or difference between two or more groups of samples (i.e., pixels). Nonetheless, prior to implementing clustering, there are various methods for calculating the similarity between pixels in a dataset. This entails the measuring of distances (similarity) between pixels whereby similar pixels are assembled in the same terminal node of a tree more often than contrasting pixels [45]. This is a procedure that is efficiently achieved through computing a random forests (RF) proximity matrix. Here, the non-linear distance between pixels is scored by counting the number of trees that used the same routes to classify them. So, the distance between similar pixels is expected to be larger than dissimilar pixels which often yield lower distance values [55]. Based on the pixels' similarity, the proximity matrix creates patterns within a dataset which affords an unsupervised automated procedure for detecting anomalous pixels and enables the clustering of unlabeled classes [45]. In our case, the distance is based on relative water content pixels, where drought-affected trees will exhibit lower water content than non-affected ones. The output is expected to shed light on how trees respond to drought-stress—a particularly imperative necessity given the recent evidence of recurrent catastrophic drought events in the Zululand forestry region of South Africa. While the response of the entire KwaMbonambi forest to intense drought conditions was reported by Xulu et al. [2], their results showed variations of trees' reaction to drought stress. Such patterns could be uncovered through proximity analysis.

For this study, we constructed a matrix of 60 months (2013–2017), incorporating 383 forest compartments in an attempt to create clusters of trees with homogeneous (non-drought affected) and non-homogeneous (potentially drought affected) water content as measured by NDWI. It should be noted that the drought started in 2014, intensified further during 2015 and ceased in 2016, and in this study, we decided to include the period before (2013) and after (2017) the event for a complete analysis. We then employed *k*-means cluster analysis based on NDWI as a drought characteristic to separate compartments into mutually exclusive drought-affected and non-drought affected clusters. *K*-means is a non-hierarchical clustering method that partitions a dataset into K clusters by minimizing the sum of squared distance in each cluster [56].

For comparison, we applied three more popular matrices for calculating distances between samples: RF proximity matrix, Manhattan and Euclidean (linear) measures. Euclidean distance computes the root square difference between the coordinates of a pair of objects, whereas Manhattan computes the absolute difference between the coordinates of a pair of objects [56]. Therefore, the distance matrices computed (RF proximity matrix, Euclidean and Manhattan) were then used in the clustering analysis which was performed using Wards Hierarchical clustering. Hierarchical clustering is an unsupervised learning technique that combines cases into homogeneous clusters by merging them into tree-like hierarchical diagrams displaying associations among all variables in a given data set [57], as illustrated in Figure 2. The y-axis represents the similarity between objects, which entails the measure of closeness of their individual data points, and the x-axis represents the objects. In this example, A is more similar to C than F, but is dissimilar to N. The overall procedure followed in this study to cluster forest compartments into drought-affected and non-affected is illustrated in Figure 3. The unsupervised RF proximity matrix was generated using the R statistical software [58] using the "randomForest" package, while clustering (hierarchical and non-hierarchical) analysis was implemented and presented using the "cluster" and "stats" packages.

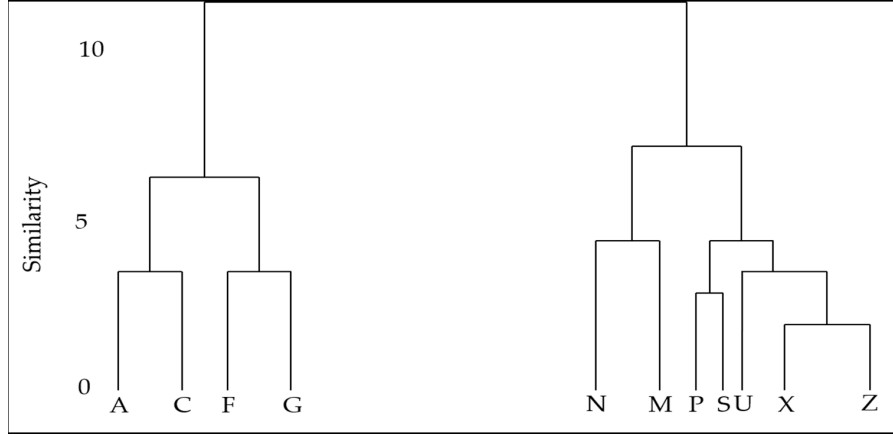

**Figure 2.** A graphical illustration of the dendrogram, where clusters at one level are joined together to form the clusters at the next levels.

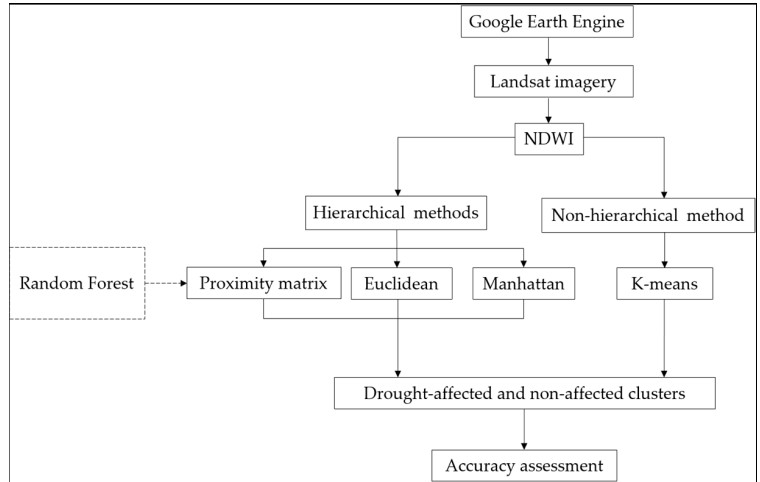

**Figure 3.** Flow diagram of drought clustering analysis.

*2.4. Accuracy Assessment*

To evaluate the performance of the unsupervised methods for detecting the anomalous NDWI values, we used the overall accuracy (OA). The OA was computed by dividing the sum of correctly classified drought entries by the total number of sampled forest compartments. We also calculated the producer's accuracy (PA), which is calculated by dividing the number of correctly classified pixels in each category by the number of reference pixels "known" to be of that category, and the user's accuracy (UA), which is computed by dividing the number of correctly classified pixels in each category by the total number of pixels that were classified in that category [59].

$$PA = \frac{A}{A + B} \times 100 \tag{2}$$

where A is the number of drought affected compartments that were correctly classified as affected by drought and B is the number of compartments that were incorrectly classified.

$$UA = \frac{C}{C + D} \times 100 \tag{3}$$

where C is the number of non-affected drought compartments that were incorrectly classified as drought affected and D is also a number of non-affected drought compartments that were classified correctly. The overall classification (OA) accuracies were determined by the following equation:

$$OA = \frac{A + D}{A + B + C + D} \times 100 \qquad (4)$$

Field data for each compartment (*n* = 383) was collected annually between 2013 and 2017 using scheduled field surveys. Trees were physically observed for drought damage using expert knowledge and were recorded on a compartment basis. The data collected at the field included the presence of damage and number of trees damaged. Only compartments displaying severe damage of greater than 50% were recorded and stored. GPS points were then used from a differentially corrected Trimble GeoXT hand held receiver with an accuracy of <2 m to match field data to compartment polygons which were extracted from the company's management database. It is worth mentioning that the RF unsupervised algorithm does not work well with unbalanced classes and, therefore, a subset of 114 compartments comprising drought-stressed (*n* = 57) and non-stressed (*n* = 57) was used to assess the classification methods, and the full set (*n* = 383) containing drought-stressed (*n* = 57) and non-stressed (*n* = 326) was only used to show the spatial patterns. For in-depth analysis, we compared the performance of the RF derived proximity matrix, Euclidean and the Manhattan distance matrices for detecting drought-affected forest compartments.

## 3. Results and Discussion

We aimed to establish whether *Eucalyptus* compartments in drought-prone Zululand could be separated into drought-affected and non-affected using cluster analysis based on the NDWI as a drought indicator and, if so, to what extent. Cluster plots were constructed using the *k*-means algorithm to separate compartments based on their water content. Figure 4 shows the outcome. We grouped 114 *Eucalyptus* forest compartments of which NDWI had similar temporal variability between 2013 and 2017. For the sake of simplicity, we considered all 57 drought-affected compartments and randomly sampled a total of 57 non-drought compartments for analysis. The results distinguished two main clusters differing in their response to drought, as illustrated in Figure 4. The combined clusters explained 57.4% of the variation in NDWI values in the data set, individually accounting for 34.1% (Dim1) and 23.3% (Dim2), respectively. Cluster 1 comprises 57 forest compartments classified as drought-affected, mainly due to their relatively low values of NDWI (0.01–0.46), and cluster 2 also contains 57 randomly sampled forest compartments classified as non-drought compartments, totaling 114 compartments. It is instructive to notice the performance of different similarity measures in separating the data. As illustrated in Figure 4, the RF proximity matrix appears to separate drought-affected from non-affected better than Manhattan and Euclidean measures. Figure 4 also shows that Euclidean has a relative overlap between the two clusters which is relatively greater than the other two. In all cases, the drought-affected cluster seems to be more uniform, whereas the non-affected compartments, on the other hand, show a relatively broad distribution. These distinct characteristics between the groups can be seen in Figure 4.

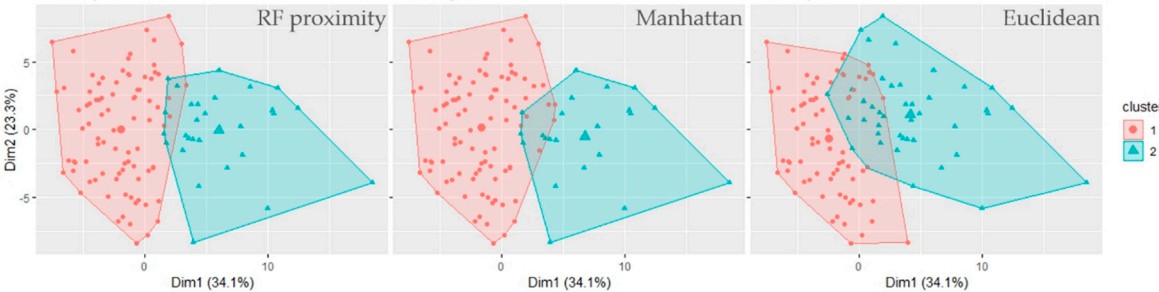

**Figure 4.** The cluster plots of three similarity measures of trees based on Normalized Difference Water Index (NDWI). Cluster 1 represents drought-affected trees and cluster 2 represents non-drought affected trees.

Having shown the separation of the two main clusters, we extend the analysis by further partitioning the clusters based on their responsiveness as measured by the NDWI. To achieve this,

hierarchical clustering was performed using Ward's method as an amalgamation rule and RF proximity as a measure of dissimilarity. The dissimilarity level of the height (difference in NDWI values) was set at 2.5, resulting in two main clusters—the drought-affected and the non-affected—as illustrated in the dendrogram in Figure 5. This is because drought-affected and non-affected clusters are clearly separable at the 2.5 similarity level.

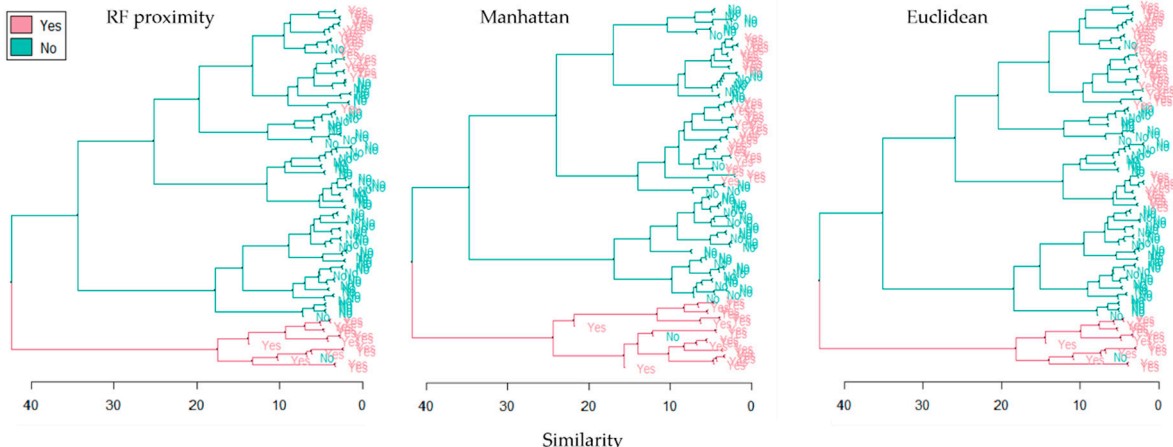

**Figure 5.** Hierarchical clustering of drought-affected (Yes) and non-affected (No) compartments using three similarity measures based on NDWI.

The first cluster consists mainly of drought-affected forest compartments and the second cluster is not affected by drought. As shown in Figure 5, the application of the RF proximity matrix was relatively better in grouping forest compartments in their appropriate clusters and the dissimilarity level of a few compartments between different clusters was too small. The performance of both Manhattan and Euclidean distance was almost similar, and the dissimilarity level of relatively more compartments between clusters was higher than the RF proximity matrix. This inconsistency seems to emphasize the combined effect of site and species physiological mechanisms since tree species [60], and even individuals [61], differ in their reaction to the 2015 drought event. Another possible source of complication is silvicultural practices such as harvesting, which would naturally affect the Landsat signal. This together with shifting climate conditions makes it difficult to adopt the means for consistent spatial and temporal monitoring of drought impacts. Despite these obstacles, the hierarchical clustering based on NDWI was able to reliably and accurately record the drought characteristics of *Eucalyptus* plantations.

Figure 5 confirms that hierarchical clustering analysis using the RF proximity matrix revealed high consistency in grouping drought-affected forest compartments; a negligible number of non-drought compartments were spotted within cluster 1. This is, in part, due to less variation in water content because trees were already suffering from water stress. Also, some drought-affected compartments were represented within a non-drought cluster because of the complication of the greater variability of water content in the trees that were represented in such compartments.

The results showed that the unsupervised RF proximity matrix is capable of separating homogenous forest compartments based on their drought characteristics with consistently high accuracies (Table 1). More specifically, the RF proximity matrix achieved the highest overall accuracy of 87.7%, followed by Manhattan (85.9%) and Euclidean (79.9%) similarity measures. The performance of the RF proximity matrix was superior in detecting drought-affected compartments and non-drought compartments, with producer's and user's accuracies ranging from 84.2% to 91.2%. Manhattan ranged from 42.8% to 98.2%, while the Euclidean similarity measure had accuracies stretching from 37.2% to 94.7%. These results confirm conclusions by Peerbhay et al. [45,62], who also showed the superiority

of using the proximity matrix to map anomalous bugweed pixels in commercial forest plantations using remotely sensed datasets.

**Table 1.** Accuracy assessments for different similarity measures (drought stressed ($n = 57$) and non-stressed ($n = 57$)).

| Similarity Measure | Producer's Accuracy (%) | | User's Accuracy (%) | | Overall Accuracy (%) |
| --- | --- | --- | --- | --- | --- |
| | Drought | No-Drought | Drought | No-Drought | |
| Random Forest (RF) proximity | 84.2 | 91.2 | 90.5 | 85.2 | 87.7 |
| Manhattan | 73.6 | 98.2 | 42.8 | 57.1 | 85.9 |
| Euclidean | 65.3 | 94.7 | 37.2 | 62.7 | 79.9 |

Time series of monthly NDWI values from 2013 to 2017 for all clusters are presented in Figure 6. In this figure, sites with trees with relatively high water content are displayed as darker colors, whereas the water-stressed ones are shown as lighter colors. The results reveal a wave-like pattern of plant water content, with notable reduction in four successive time slices: (a) August 2013 to December 2014, (b) August 2014 to February 2015, (c) August 2015 to March 2016 and (d) August 2016 to February 2017. This implies that there were short breaks in the drought for both clusters, with marked seasonal cycles of drought severity over the study period. This validates Baudoin's [15] assertion that the 2015 drought was the longest of all drought events in the South African record, especially at two successive seasonal scales (2014–2016) and was of profound magnitude. These effects were more prominent in the drought-affected cluster (cluster 1); this corresponds to the light elements in the heat map, which reveals distinct tonal differences for both clusters 1 and 2. There is a markedly darker tone for the non-drought cluster, especially from March to July each year, and the consistently lighter tone for the drought-affected cluster, respectively displaying higher-to-medium and low NDWI values. Disparities between these clusters in terms of NDWI variability are evident, such that the least variation is observed in the drought-affected cluster of trees, whereas the most effect corresponds to the greatest influence of drought.

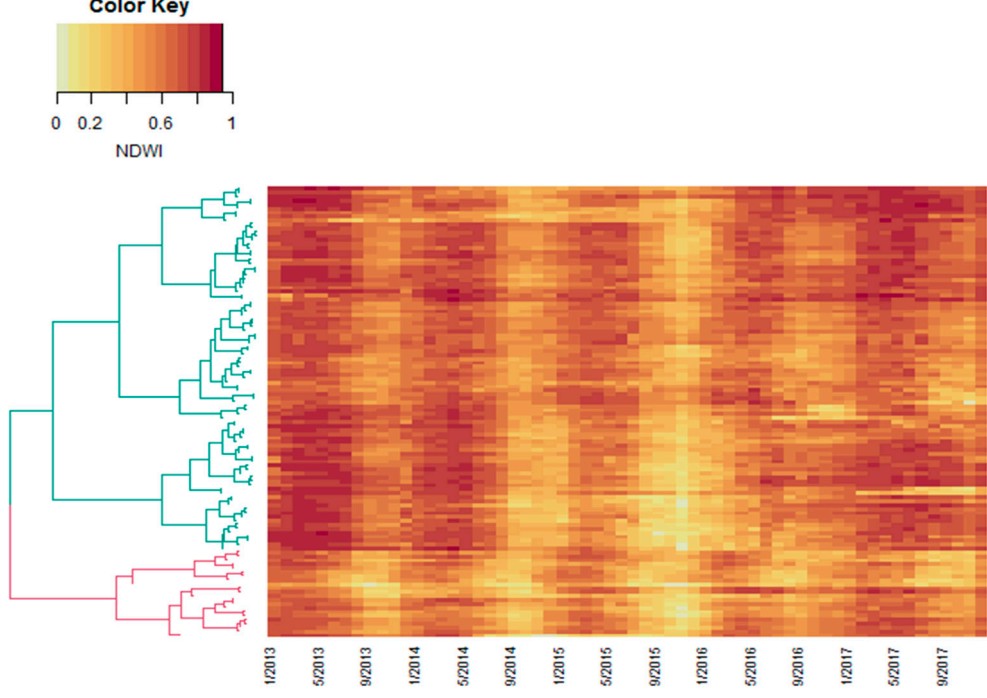

**Figure 6.** Figure showing the variation of NDWI values for each cluster from 2013 to 2017. Darker tones represent high NDWI values and light tones represent low NDWI values. Red in the classification tree represents drought-affected compartments and blue represents non-affected compartments.

The greatest decline in the NDWI values was observed for November 2015, when the lighter tones in the heat map coincide with the drought-affected cluster. This area received a relatively low monthly mean (48.8 mm) rainfall during this quarter compared to other years (Figure 7). The effect of the drought in November is also noticeable in the non-drought cluster, implying that even the drought-tolerant hybrid trees were affected at that time. This exceptional reduction in vegetation water content coincides with the combined effects of the most extreme El Niño event [15] and record-breaking temperatures [63]. This El Niño caused a delay in the onset of rainfall and a decline in the total precipitation in the summer rainfall areas of South Africa, with its associated water stress for vegetation [64], with an exceptional impact on Zululand where our study was based. This observation may explain widespread drought-induced tree dieback reported by Crous et al. [3] in this area. These results are in harmony with those of Xulu et al. [2], who observed a substantial reduction of forest canopy over the entire KwaMbonambi plantation in 2015. These authors further noted different reactions of trees in different parts of the plantation, where most trees exhibited lower NDVI values while some remnants remained relatively stable; the current study offers some explanations in that some trees are drought-responsive while others are not. Given the instructive insights offered by heat map analysis pertaining to the changes in water content of vegetation in each cluster over the study period, the question that arises now is, in which part of the plantation are the changes identified apparent?

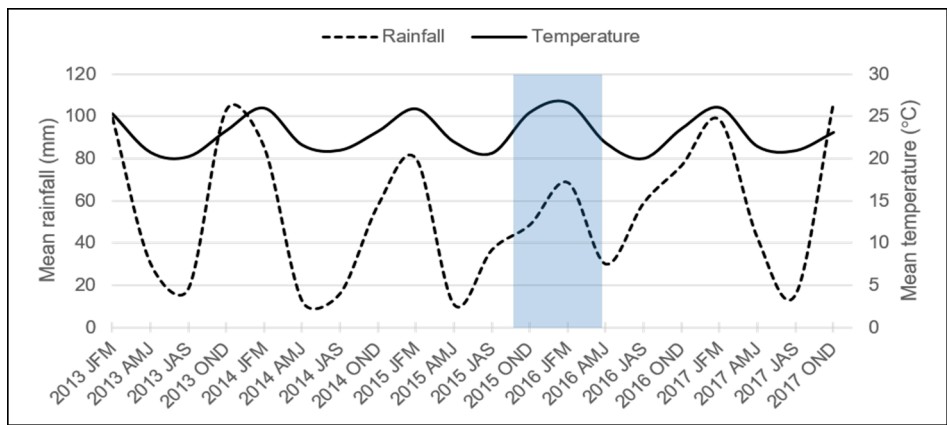

**Figure 7.** Quarterly mean rainfall and temperature over KwaMbonambi from 2013 to 2017.

After clustering the drought-affected and non-affected forest compartments, we constructed quarterly averaged maps of the spatiotemporal patterns of NDWI over the area studied between 2013 and 2017 (Figure 8), which indicates the consistency with the results in Figure 6. Spatially, the results display a patchy distribution of low NDWI values (shown as red and orange) across the study area, but appear relatively prominent in the central-east towards the northern part of the plantation. This northern part, towards the town of Mtubatuba, experiences long-lasting, drier conditions and hence it is expected that trees over this section should exhibit lower values of NDWI. In fact, the SA Forestry [4] reported a greater decline of growth trends over drier areas north of Mtubatuba, where an ominous 35% to 40% reduction in the *E.g×u* clone was apparent during the drought period. Unfortunately, the current *E. g×u* clone is maladapted to extreme drought conditions, and the *Corymbia* hybrids are supported in this area of extreme drought which prevailed from 2014 to 2016. Generally, from 2013 to 2015, the results show a fluctuating but decreasing tendency in NDWI values when water stress in the plantation was widespread. During this period, South Africa recorded the lowest annual rainfall since 1904 [65]; it is assumed to have been worse over the drought-prone region studied. A slight recovery in rainfall receipts is noticeable from 2016, particularly in the central parts of the study area towards the north.

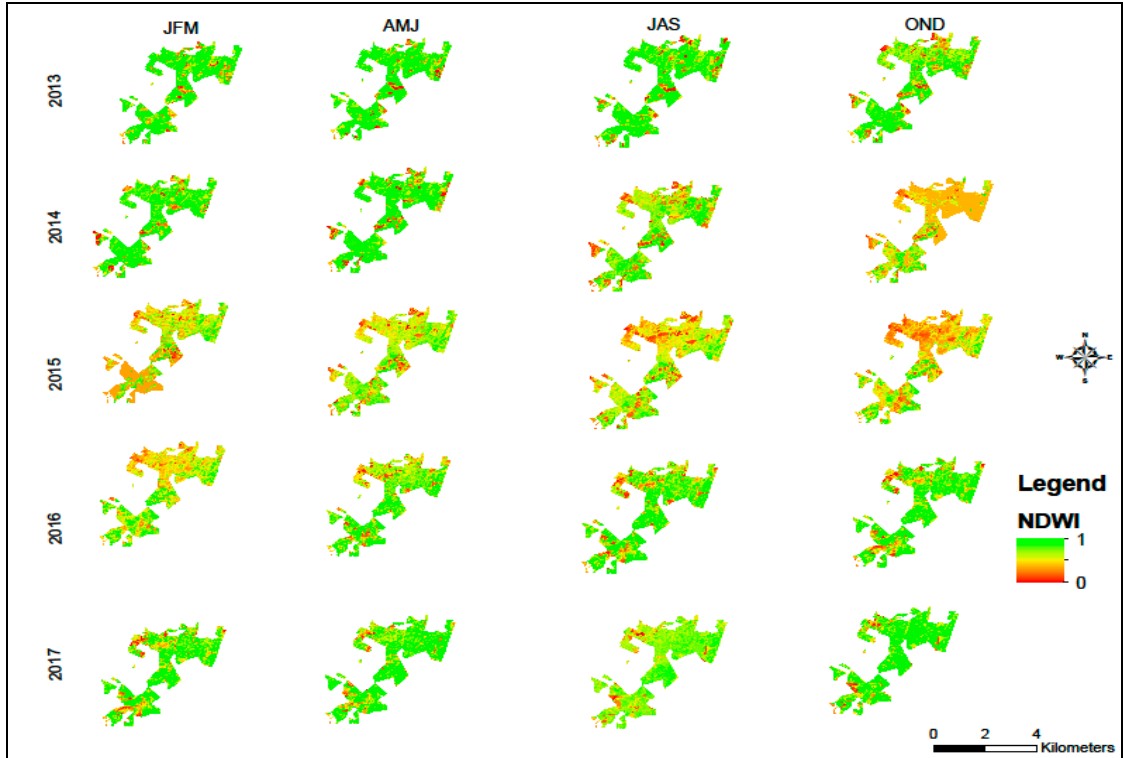

**Figure 8.** The spatial pattern of quarterly NDWI values over KwaMbonambi from 2013 to 2017.

Overall, our results demonstrate the value of Google Earth Engine and the RF proximity matrix for proving to be a cost-effective and fast approach to discriminate drought-affected trees from non-affected ones. This is particularly relevant for countries such as South Africa that are frequently affected by drought events and of which their economy is largely dependent on agricultural systems. We hope our results can help to inform drought management plans so that forest productivity is optimized in the face of a shifting climate.

## 4. Conclusions

Our results demonstrate that hierarchical cluster analysis is a practical approach to group and delineate forest plantations based on their drought characteristics. The RF proximity matrix successfully identified drought-affected from non-affected forest compartments with high overall accuracy of 87.7%, surpassing Manhattan (85.9%) and Euclidean (79.9%) distance measures. Cluster 1 displayed a uniform pattern of and mainly lower NDWI values. Cluster 2 showed more variation in NDWI values. We found a few overlapping clusters, partly due to the varying sensitivity of hybrid *Eucalyptus* clones to drought stress, which presented complications for clustering. Furthermore, the heat map allowed us to determine the temporal evolution of water stress in the clusters. We observed the apparent reduction in leaf water content in all the years studied, starting in the summer, with the most substantial decline in November 2015, which coincided with an extreme El Niño event and the hottest period in the climate records. Spatially, our results showed furthermore that the recent intense drought had a differential impact on the KwaMbonambi plantation, which was particularly pronounced over the central-east towards the northern part of the forest. Overall, unsupervised hierarchical clustering analysis indicate that the NDWI can explain the patterns of drought-stress in homogeneous planted forests with high fidelity and with more insight in our study area than had previously. Future research may be extended to broader scales and may also include other species, particularly in areas affected by drier drought conditions. It would also be ideal to test the model of different forests so as to identify factors that could improve its performance. Clear-cut compartments may present challenges in the performance of the method, since the water content is greatly altered during the process. Hierarchical

cluster analysis is a superior method for improving our understanding of the complex response of planted forests to drought stress for these reasons: (a) it can handle large amounts of data and is able to reach conclusions rapidly; (b) dendrograms and heat maps display the associations of the tree species involved without user prejudice; (c) the R software provides unrestricted state-of-the-art statistical analysis and visualization and (d) the full range of high-density time series data for vegetation is readily available on cloud-based high-performance computing systems.

**Author Contributions:** Conceptualization, S.X., R.I., K.P. and M.G.; methodology, formal analysis, resources, writing of original draft, S.X., R.I. and K.P. and review and editing of final manuscript: S.X., R.I., K.P. and M.G.

**Funding:** This research was funded by the South African National Space Agency (SANSA), the University of KwaZulu-Natal and the National Research Foundation (NRF) of South Africa (grant number 114898).

**Acknowledgments:** The authors would like to thank Sappi Forests-SA for granting access to the study sites and excellent working conditions.

**Conflicts of Interest:** The authors declare no conflict of interest.

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
