# Peer review of "Unsupervised Clustering of Forest Response to Drought Stress in Zululand Region, South Africa"

_forests, doi:10.3390/f10070531_

Round 1

Reviewer 1 Report

The manuscript has been considerably improved from the previous version and most of the comments addressed.

After reviewing the updated manuscript, I have the following minor comments:

·         GEE needs to be defined first time used in the main text. This is, add the abbreviation into parenthesis in LN129.

·         LN 37      GDP is only used once in the main text, therefore, the abbreviation is not needed

·         LN 37     R31 - net revenue? : This needs to be defined in the text. Also, consider that in LN102 the letter R is used for red (R) and LN391 R software

·         LN 37     USD 918 092 058 – add comma/point

·         LN 38     employs more than 200 000 of the national labour force – add comma

·         LN 38      (DAFF 2017) – is this a reference? add definition

·         LN 41     “For example, downward growth patterns were highest during the recent drought” – specify the drought you refer to (2015?)

·         LN 60     “During this period, the country received the record-setting (403 mm) annual rainfall” – add the historical mean annual rainfall shown in LN144 (so the 403mm value can be clearly seen as a very low value in comparison with the mean value)

·         LN 72     “resulting in extensive tree dieback” – please quantify “extensive”, i.e. number of trees

·         LN 95-97: “simplified drought index (i.e. standardized precipitation index (SPI)); a single quantitative value that incorporates a great amount of environmental data” – this sentence is erroneous. The SPI index is only based the long-term precipitation record (refer to the sentence in LN114-116).  Therefore, the sentence “that incorporates a great amount of environmental data” is wrong.

·         LN98      it is suggested to change “strength” for “magnitude and intensity”

·         LN144   “The annual rainfall varies from 739 to 1219” – I assume this should be read as “MEAN”

·         LN146   “the potential 145 evapotranspiration is commonly in the range of 1100 to 1772 mm” – I assume this should be read as “MEAN”

·         LN169   The manuscript needs to explain the reason(s) why such a short time span 2013-2017 of analysis has been used (i.e. as per the author’s response: “Longer time period like 10 years ideal, but in our case, the analysis informed by the Sappi inventory data spans from 2013. In our next series of papers, we will consider the longer period as suggested”) and  mention possible consequences/shortcoming/limitation of this

·         LN208   “In this example, A is similar to C than F but dissimilar to N” – “more” missing? A is more similar to C than F?

Author Response

The manuscript has been considerably improved from the previous version and most of the comments addressed. After reviewing the updated manuscript, I have the following minor comments:

GEE needs to be defined first time used in the main text. This is, add the abbreviation into parenthesis in LN129.

The abbreviation GEE was added in Line 134

LN 37      GDP is only used once in the main text, therefore, the abbreviation is not needed

The GDP in Line 38 was deleted as was suggested

LN 37     R31 - net revenue? : This needs to be defined in the text. Also, consider that in LN102 the letter R is used for red (R) and LN391 R software

Net revenue was added in the sentence in Line 36–37 and it now reads “Commercial forestry sector plays an important role in the economic development of South Africa as it contributes a net revenue of…” R for red in Line 102, now Line 105 was deleted, and the only R is for R software in Line 403

LN 37     USD 918 092 058 – add comma/point

The comma was added in Line 37 and the value is now USD 918, 092, 058.

LN 38     employs more than 200 000 of the national labour force – add comma

The comma was added in Line 38 and the value is now 200, 000

LN 38      (DAFF 2017) – is this a reference? add definition

The reference for (DAFF 2017) is now [1] in Line 38, and the reference list was updated accordingly.

LN 41    “For example, downward growth patterns were highest during the recent drought” – specify the drought you refer to (2015?)

The recent drought has now been specified by adding 2015 in Line 42. Now the sentence reads “…. recent 2015 drought…”

LN 60     “During this period, the country received the record-setting (403 mm) annual rainfall” – add the historical mean annual rainfall shown in LN144 (so the 403mm value can be clearly seen as a very low value in comparison with the mean value)

The historical average annual rainfall was added in Line 62-63. …“a reference to the record-setting 403 mm rainfall. The sentence in Line 61-63 now reads  “During this period, the country received the record-setting (403 mm) annual rainfall since the South African Weather Service began collecting rainfall data in 1904 [17]; a record lower than the historical mean annual rainfall of 739 mm.

LN 72     “resulting in extensive tree dieback” – please quantify “extensive”, i.e. number of trees

The number of trees were not quantified in the above study as the paper assessed the overall impact of drought over the health of Eucalyptus forest compartments which displayed major or ‘extensive’ negative responses during the drought period. The word “extensive” was therefore used to explain the seriousness of the impact of drought on the studies forest resources.

LN 95-97: “simplified drought index (i.e. standardized precipitation index (SPI)); a single quantitative value that incorporates a great amount of environmental data” – this sentence is erroneous. The SPI index is only based the long-term precipitation record (refer to the sentence in LN114-116).  Therefore, the sentence “that incorporates a great amount of environmental data” is wrong.

The sentence “…that incorporates a great amount of environmental data…” in Line 100 was deleted

LN98      it is suggested to change “strength” for “magnitude and intensity”

“strength was replaced by magnitude and intensity” in Line 101 as suggested

LN144   “The annual rainfall varies from 739 to 1219” – I assume this should be read as “MEAN”

The word “mean” was added in Line 149

LN146   “the potential 145 evapotranspiration is commonly in the range of 1100 to 1772 mm” – I assume this should be read as “MEAN”

The word “mean” was added in Line 151

LN169   The manuscript needs to explain the reason(s) why such a short time span 2013-2017 of analysis has been used (i.e. as per the author’s response: “Longer time period like 10 years ideal, but in our case, the analysis informed by the Sappi inventory data spans from 2013. In our next series of papers, we will consider the longer period as suggested”) and  mention possible consequences/shortcoming/limitation of this

As mentioned on page 7 lines 225-261, during this time period (2013-2017) accurate ground data was available for this study region. Since the drought began in 2014, intensified during 2015 and ceased in 2016, the field data coincided with this event. Nonetheless, we also included data for a year before 2014 and a year after 2016 since this data was also collected and available (Page 5 lines 205-208). While longer periods such as 10 years are ideal, the dataset in this study accurately captures the drought event in this region and the impact on forest resources thus provides a platform for ongoing drought monitoring work in this area and over longer periods of time.

Some possible consequences and short comings of this study have been mentioned in the discussion section on page 2 Lines 76-78, whereby repeated monitoring for drought over larger time frames (i.e. > 10 years) would uncover time series patterns and aid in early detection efforts for strategic forest protection management decisions. Nonetheless, this study demonstrates the effectiveness of detecting the impact of drought using an unsupervised clustering framework for repeatable and reliable mapping results. (Page 12 Lines 247-433)

LN208   “In this example, A is similar to C than F but dissimilar to N” – “more” missing? A is more similar to C than F?

“more” was added in Line 222

Reviewer 2 Report

Dear Authors,

I am not satisfied by the responses and improvements I have received. When a major revision is required, I expect a response to all my comments with a thorough explanation of the changes that were made, where and why. 

Attached some additional comments.

Good luck with getting this work published.

Author Response

Comment 1:  Unfortunately, the in Line 139

“Unfortunately, the in Line 139” - This was deleted in the text as indicated. The sentence now starts with E.gxu in Line 144.

Comment 2:  Add “more” in Line 139

“…more…” was added in Line 144 as suggested. The sentence now reads “E.gxu is more…”

Comment 3:  The size of the ‘compartments’ in ha still not included

The size of compartments was added in Line 182. The sentence now reads “…study area of almost 20, 000 ha were…”

Comment 4:  Explain in your caption what the y-axis means (similarity).

The description of both y- and x-axis were added. The new sentence in Line 220 reads “The y-axis represents the similarity between objects which entails the measure of closeness of their individual data points, and the x-axis represents the objects. ”

Comment 5:  What is Sappi’s ground data, how and when was this data collected?

Sappi ground data has now been explained. “Field data for each compartment (n = 383) was collected annually between 2013 and 2017 using scheduled field surveys. Trees were physically observed for drought damage using expert knowledge and recorded on a compartment basis. The data collected at the field included presence of damage and number of trees damaged. Compartments only displaying severe damage of greater than 50% were only recorded and stored. GPS points were then used from a differentially corrected Trimble GeoXT hand held receiver with an accuracy of < 2m to match field data to compartment polygons extracted from the company’s management database” in Line 255-265.

Comment 6:  Addition of x 100 to get % as reported in Table 1

The formula for PA, UA and OA (Equation 2 – 4) were updated by adding X 100 to get % as reported in Table 1. This is in after 248-254.

Comment 7:  Proportion of number in Line 229

“Proportion” was changed to “number”

Comment 8:  You calculated these for affected and non-affected sites (as in Table 1). This means that A, B, C, D should be different when you calculate these indexes for affected or non-affected…

This is correct. A and B is used specifically to calculate producer’s accuracies (PA) and C and D is utilized specifically for calculating user’s accuracy (UA). As displayed in Table 1, both PA and UA are different for affected and non-affected compartments.

Table 1

Similarity measure 

Producer’s accuracy (%)

User’s accuracy (%)

Overall accuracy (%)

Drought

No-drought

Drought

No-drought

RF proximity

84.2

91.2

90.5

85.2

87.7

Manhattan

73.6

98.2

42.8

57.1

85.9

Euclidean

65.3

94.7

37.2

62.7

79.9

Comment 9:  How the Proximity matrix, Euclidean, and Manhattan were compared?

The performance of Proximity matrix, Euclidean, and Manhattan for detecting drought and non-affected compartments were compared in Figure 4, and quantitatively in Table 1. The User, Producer and Overall accuracies were also reported in the results.

Comment 10:  What (statistical) packages were used for the analysis. How were the statistics presented further down, generated? Explain

This is now explained in Line 244. “The unsupervised RF proximity matrix was generated using the R statistical software (R Development Core Team, 2012) using the ‘randomForest’ package while clustering (hierarchical and non-hierarchical) analysis was implemented and presented using the ‘cluster’ and ‘stats’ packages”

Comment 11:  You changed the sample sizes, but the values and figures in your manuscript have not changed. Please explain how this is possible? I would assume that a different sample size would give you different outcomes…

The sample size for the analysis has not changed, hence the figures in Table 1. The clarity on the sample size pointed by reviewers in the initial version of the manuscript made us to rectify the figures that were not specified in text. The sample remains a total of 114 compartments, half of which drought-affected and the other is non-affected.

Comment 12:  Describe here what a, and b stand for. How are they different?

In Figure 5, a, and b is basically the same, and b has been deleted to remain with only one figure – the dendrogram showing the all three distance measures and also indicated in comment 13 below.

Comment 13:  From your text, it seems this (Figure 5) was only done for Euclidean distance. Why not for all?

In line with the previous comment, Figure 5 has now been amended to include the dendrograms for the RF proximity matrix and the Manhattan analysis.

Comment 14:  The shading is now considerably different from the original manuscript. Is this due to the different samples size used? Needs explaining in thorough response.

The sample size for the analysis has not changed. In the initial manuscript, the output was saved as JPG and the latest one is PDF to enhance the qu
